# The Work Ability of People with Mental Illnesses: A Conceptual Analysis

**DOI:** 10.3390/ijerph181910172

**Published:** 2021-09-28

**Authors:** Yoshitomo Fukuura, Yukako Shigematsu

**Affiliations:** 1Department of Nursing, Graduate School of Medicine, Kurume University, Kurume 830-0003, Fukuoka-ken, Japan; 2Department of Nursing, School of Nursing, Kurume University, Kurume 830-0003, Fukuoka-ken, Japan; shigematsu_yukako@med.kurume-u.ac.jp

**Keywords:** work ability, mental illness, employment, work environment, mental healthcare, rehabilitation, self-management, motivation, performance

## Abstract

Continuous employment is an important goal for many people with mental illnesses. However, job retention and absenteeism remain significant problems that deter their attempts at gaining financial independence, expanding interpersonal relationships, and developing self-esteem. Although there is consensus on the necessity of their proactive treatment and social participation, such support is currently underwhelming. Therefore, this study analyzes the conceptual framework for work ability of people with mental illnesses. We performed our investigations using Rodgers’ evolutionary conceptual analysis, targeting literature published from 1978 to 2020. Our search yielded 1420 articles in the Scopus inventory and 199 in PubMed. After exclusions, 13 articles remained. Using the same keywords in Google Scholar, we included 31 articles in our analysis. The attributes of work ability included the ability to self-manage, adaptability, the ability to dedicate oneself to work, and the ability to formulate plans. These were developed through a reiterative process. This study notes the importance of adjusting the work environment according to the patients’ condition. Therefore, the ability to cope with stress and workload, as well as active self-adjustment, are crucial skills that nurses can help develop after assessing the patient’s daily life. Furthermore, they can foster multidisciplinary collaboration and follow-up systems after employment.

## 1. Introduction

Working is an important goal for people with mental illnesses. However, job retention and absenteeism remain a significant problem [1,2,3]. These factors deter people with mental illnesses from attempting to gain financial independence, expand interpersonal relationships, and develop self-esteem. Ultimately, this results in a decline in psychosocial functioning [4].

To address these issues, the American Psychiatric Nursing Association recommends the provision of support toward proactive treatment and social participation [5]. Furthermore, mental health services also emphasize a holistic approach rather than the treatment of symptoms. Professionals are required to introduce new interventions consisting of recovery and positive psychology [6]. Among them, psychiatric nurses should comprehensively plan psychosocial rehabilitation, symptomatology management, the promotion of social skills, and interventions for cognitive improvement [7]. Therefore, psychiatric nursing emphasizes the need for overall “well-being” when striving for achievement through valuable social roles or participation in the workplace [5].

However, nursing for adults with disabilities also highlights the value of employment support as a goal of social participation. In clinical practice, there are protocols that run parallel to job searching, but nursing has yet to be organized in preparation for employment. Studies on the employment of mentally handicapped people, in the fields of social welfare and rehabilitation, have aided in the gradual systematization of approaches for achieving competitive employment. Individual Placement and Support (IPS) exemplifies this approach. Although IPS has produced desirable results, the lack of a measurement framework limits their generalizability [8]. Furthermore, a framework exists for work-related social skills, applicable to psychiatric rehabilitation and consisting of three stages: basic skills, core skills, and achievements. However, the framework remains unclear and inconsistent [9].

At present, in terms of national policy, the United Kingdom has enacted rational adjustments that allow workers with disabilities or health conditions to secure working places during their rehabilitation [10]. Authorities in the United States have developed soft skills training, emphasizing collaboration and problem-solving [11]. In Japan, there are structures that develop basic skills, such as critical thinking, for working adults [12]. However, although there is consensus on the necessity of support for the employment of persons with mental illnesses, such support is underwhelming in its current iteration. As psychiatric nursing aims to support medical and social life, it is necessary to clarify its role in supporting the employment of persons with mental disorders. This clarification will inform the understanding of the work ability of this population, as well as the factors related to it. Doing so would allow us to examine the role of nursing for employment, specific support, and evaluative indices.

Therefore, this study aims to analyze the conceptual framework of the work ability of people with mental illnesses.

## 2. Materials and Methods

We performed our investigations using Rodgers’ evolutionary conceptual analysis [13]. This qualitative and inductive method clarifies the concept of interest; collects a wide range of literature data; and clarifies the attributes, their antecedents, and their consequences. Specifically, the attributes indicate the characteristics of the concept’s utility, its antecedents occur prior to the attributes, and the consequences occur because of the attribute. This method focuses on the qualities and usage of the concept as an entity that evolves according to the given context. Thus, we believe that Rodgers’ conceptual analysis is applicable to the examination of the work ability of persons with mental disorders, which is greatly dependent on historical, cultural, and ethical contexts.

This study targeted literature published from 1978 to 2020, to visualize the concept of “work ability of persons with mental disorders”. The academic fields used for data collection included nursing, psychiatry, sociology, welfare, and psychology, while the searched inventories included Scopus, PubMed, and Google Scholar. We searched for literature bearing the words “mental”, “employment/work”, “ability”, “motivation”, and “performance” as keywords. The search yielded 1420 articles in Scopus and 199 articles in PubMed. We confirmed the titles and abstracts, and excluded studies targeting diseases not specific to older adults, students, children, and people with mental illnesses aged 65 and over. Of these, 13 articles in Scopus and six articles in PubMed were found. All six PubMed articles appeared in the Scopus results. Using the same keywords in Google Scholar, we examined the titles that were perceived to explore the usage of the aforementioned concept and searched 30 articles. This was because Rodgers’ conceptual analysis collects and analyzes about 20% of the collected target literature, or at least 30 papers as a sample. Ultimately, this study included 31 pieces of literature in the analysis (Figure 1).

During our analysis, raw data about the attributes, antecedents, and consequences that appeared to correspond to our context were extracted from each piece of literature. The data were labeled, and codes were created. We created names to indicate sub-categories while examining any disparities among the codes. Any further abstracted codes were considered categories.

## 3. Results

We analyzed 31 articles in this study: 12 from the United States, 3 from Asia, and 16 from Europe. Depending on the field of the journal, we assigned the pieces of literature to the following five categories: mental health (*n* = 2), rehabilitation (*n* = 6), psychology (*n* = 8), occupational medicine (*n* = 5), and psychiatry (*n* = 10). The conceptual analysis yielded 4 attributes, 7 antecedents, and 5 consequences of the “work ability of people with mental illnesses”.

### 3.1. Attributes of Work Ability

We identified the following four attributes of the work ability of people with mental illnesses: the ability to self-manage, adaptability, the ability to dedicate oneself to work, and the ability to formulate plans.

First, we formed the ability to self-manage, according to the subject’s approach to their work. This includes “avoiding stimulation that interferes with work” [9,14,15,16,17], “self-management to avoid fatigue at work” [14,17,18], and “reviewing one’s approach toward work” [14]. The attribute that concerns “avoiding stimulation that interferes with work” includes methods for paying attention to changes in one’s physical environment and avoiding distracting places and people. “Self-management to avoid fatigue at work” includes routines done in public or private. “Reviewing one’s approach toward work” includes methods of retrospection, as well as adjusting work to prevent the aggravation of symptoms.

The attribute that concerns the subject’s adaptability involves not only the expression of one’s intention but also coordination with their surroundings. This includes “adjusting pace according to the work context” [14,19,20,21], “autonomously deciding the work procedure” [14,18,19,22,23], “collaboration with coworkers” [9,14,18,20,21,22,24], and “requesting adjustments to facilitate the continuation of work” [9,14,17,20]. “Adjusting pace according to the work context” includes adapting to the rules of the workplace. “Autonomously deciding the work procedure” includes their decision-making. “Collaboration with coworkers” includes performing tasks while communicating with colleagues and superiors. “Requesting adjustments for continuation of work” includes requesting schedule changes and cooperation.

We formed the attribute concerning the ability to dedicate oneself to work according to the skills necessary to improve work performance. This includes “maximizing the work at hand” [14,19,21,23,25,26], “paying attention to the work conditions” [14,19,26], and “conducting self-assessment” [18,21,27,28].

The ability to formulate plans concerns the approach to task completion, from preparation to the achievement of goals. This includes “considering the set-up of work procedures and methods” [14,20,21,23,24,28,29], “constructing one’s own work goals” [19,22,25,30,31], and “performing self-assessment for work” [20,21,25,28].

### 3.2. Definition of Work Ability

In terms of the employment of persons with mental illnesses, there are two definitions of “work ability”. The definition by Nordenfelt [32] uses the words “basic generic competence” to summarize the ability to collaborate, plan proactively, solve problems, and find and use information.

Meanwhile, Tengland [29] defines “work ability” as a person’s potential to “achieve quality work goals, with the associated skills required for manual, intellectual, and social dexterity”.

Based on these facts and attributes identified in this study, we defined the “work ability of persons with mental illnesses” as their ability to work and dedicate themselves to the job, in cognizance of their mental symptoms and the surrounding environment, while performing planned actions based on proactive self-assessment.

### 3.3. Antecedents of Work Ability

Certain conditions predict the work ability of persons with mental disorders. We divided the characteristics of these conditions into two categories: workplace conditions and individual abilities. Workplace conditions were further divided into three sub-categories: the workplace that believes in the employees’ ability to work, the workplace that has a support system, and the workplace with a sense of security. Similarly, individual abilities were divided into four sub-categories: skills for interpersonal communication, active orientation toward work, ability to manage health, and ability to manage basic daily living.

The workplace that believes in the employees’ ability to work is a workplace whose superiors provide constructive feedback [15,23,25,33] and support decision-making [23]. This support system enables workers to receive support from superiors or coworkers (e.g., scheduling or prioritization) [14,23,27,34]; technical support [15,23,35]; and personal guidance, if necessary [23,26,34]. A workplace with a sense of security understands employees’ illnesses and gives constructive feedback [22,27,30,36,37,38], accepts people with disabilities, and allows them to work according to their symptoms [17,37,39], while fostering a desirable workplace orientation [18,30,39].

Skills for interpersonal communication are those that facilitate “harmonious interaction” through effective communication [15,27,29,30]. The sub-category also includes the “desire for interaction” [34], “desire for recognition from one’s peers” [18,29,34], and “mutual interaction with family members”, which is indicative of family support [34,39]. Active orientation toward work includes “orientation toward a higher goal” [22,29,30,34], “attitude of seeking personal growth” [18,34], “motivation for work” [20,23,25,34,39], and “intention to utilize one’s knowledge and skills” [18,19,20,22,29,34,39]. The ability to manage health includes “symptom management required to maintain daily life”, which is premised on the management of symptoms through timeous medication and utilizing social resources [14,15,17,18,20,25,31,33,39,40,41]. This sub-category also includes the incorporation of conscious activities to stabilize symptoms [14,18,36]. The ability for basic daily living includes “cognitive abilities” such as creativity and concentration [29,41,42], “patience” [20,29], and “discipline” [16,29]. This sub-category also involves “independence with the basic activities for daily living (ADLs)”, that is, their ability to perform basic actions such as reading, writing, and speaking [20,29,39], and their “attitude toward self-evaluation”, encompassing the corrections of one’s mistakes and management through continuous reflection and accountability [14,28,29].

### 3.4. Consequences of Work Ability

Persons with mental disorders tend to attain independence, establish human connections, maintain and improve self-care, achieve happiness, and initiate positive self-transformation when they continue working.

Attaining independence includes “exercising autonomy” that allows workers to make significant life decisions [14,16,17,23,43], “attaining economic independence” from family or public benefits [16,22,43], and “recognizing one’s role” in the local community [24,26]. Establishing human connections includes “expanding interpersonal relationships”, which includes the construction of dynamic work relationships with others [15,22,26,43] and “healthy interaction with family members” [26]. Maintaining and improving self-care includes “stabilizing symptoms” and coping with their disabilities in the work environment [16,17,18,31,43], while “maintaining a healthy rhythm of life”, such as eating regular meals and waking up in the morning [16,26,43]. Achieving happiness includes “building personal interests”, such as gaining personal identity, self-esteem, and cognizance of social contracts, and utilizing social resources [18,30].

Moreover, “building self-esteem” includes the recognition of one’s intrinsic value, responsibility, and awareness of work as community participation rather than a means to earn a living [16,26,30,43]. The attribute of “acquiring abundance” involves the motivation to work at achieving goals [16,18], while the “awareness of joy in life” concerns the enjoyment of work and viewing it as, at least in part, fulfilling one’s purpose in life [16,22,34,39,43]. In this same vein, the “authentic feeling of praise” from one’s peers can also be the driving force for diligent work [26]. Positive self-transformation involves “the affirmation of identity”, which is the ability to express oneself in thought and deed [26,30]. It also involves “changing one’s self-perception and understanding others” and understanding the differences between oneself and one’s environment [16,26]. Lastly, “improving individual motivation” relates to seeking interaction and communication through work [23,26], and “improving self-efficacy” relates to positive changes through internal and external factors [9,16,23,26].

## 4. Discussion

In this study, we performed a conceptual analysis, which clarified the definitions, attributes, antecedents, and consequences that constructed the conceptual framework of the work ability of persons with mental illnesses. Thereafter, it could inform psychiatric nurses in their consideration of the characteristics of employability for members of this population to ascertain specific support and evaluative indices, as well as the significance of intervention. This study considered the skills required for persons with mental illnesses to continue working, as well as the role of nursing that supports these skills among this population.

### 4.1. The Required Work Ability for Persons with Mental Illnesses

The required skills of persons with mental illnesses are the ability to self-manage, adaptability, the ability to dedicate oneself to work, and the ability to formulate plans. In the employment of persons with mental illnesses, the most important concern for nursing is the adjustment of the associated content and environment according to the person’s medical condition. The ability to self-manage and the adaptability identified in this study are fundamental skills for the initial stage and continuation of employment for persons with mental illnesses. Potential negative characteristics include the instability of mental symptoms and difficulties in interpersonal relationships, leading to an exacerbation of symptoms due to specific stresses [44]. Therefore, meaningful employment depends on symptom management, the correct work approach, and minimizing influences on task completion. In other words, the ability to self-manage is necessary when coping with stress in a work environment. Furthermore, a productive rapport with coworkers requires adaptability. It is necessary to actively adapt to the surroundings and work environment, in addition to one’s own volition.

Thus, stress-coping and active self-adjustment form the basis of the ability to self-manage and be adaptable. These skills overlap with the antecedent ability to manage health, skills for interpersonal interaction, and active orientation toward work. Therefore, during skill development for prospective employment, it is necessary to support this fundamental ability for the person to cope with the associated stress and their environment. The ability to self-manage and adapt is an indispensable requirement since work is essentially an activity done for others [45]. Thus, we posit that if one is to have fruitful interactions with their coworkers, they must foster workplace cooperation by adjusting to the work environment.

The establishment of the ability to self-manage and be adaptable facilitates the ability to dedicate oneself, formulate plans, and prioritize competent and resourceful completion of various work tasks. Workers must devote ample energy to performing their duties, by experiencing the plan–do–act–check cycle to achieve their goals, for the benefit of the workplace and their own independence and growth. Therefore, it is necessary to work by seeking to achieve self-actualization, rather than by limiting the experience to self-adjustment and stress-coping. However, people with mental illnesses may encounter challenges when trying to concentrate or work systematically [46]. Therefore, it is necessary to cultivate concentration skills and work while performing self-adjustments, so that one can rest whenever necessary. Furthermore, the inability to work efficiently can affect one’s symptoms. This possibility necessitates the “ability to formulate plans” [47].

With this in mind, performance abilities (e.g., the ability to dedicate oneself to work and the ability to formulate plans) facilitate resourceful work performance and efficient planning to achieve professional goals. These skills tend to accelerate outcomes, such as acquiring independence, maintaining and improving self-care, achieving happiness, and positive self-transformation. Therefore, it is essential to develop these work performance skills as a means of strengthening work efficiency and planning.

For continued employment, one must consolidate one’s ability to self-manage and adapt. Meanwhile, one should develop one’s ability to dedicate oneself to work and formulate plans with an outlook toward the future. We believe that this enables one to demonstrate their abilities more fully, leading to continuous employment. Similarly, the experience of adjusting oneself to the work, gradually increasing one’s workload, and making appropriate compromises are essential for developing the abilities that constitute work ability, in addition to stress management prior to and during employment. Dunn [48] also states that acquiring new work-related skills through practical application is a vital condition for continued employment.

However, people with mental illnesses do not always develop these four skills unilaterally, due to their symptoms. Thus, even if they develop their ability to manage stress and self-adjust, their ability to self-manage and adapt may decrease with certain influences that come with the work. Therefore, coworkers, family members, outpatient nurses, and those in the workplace should assess the ongoing situation while understanding the condition of the person concerned. The antecedents necessary for maintaining employment should be considered, such as through reviewing daily life and utilizing medical services in outpatient clinics.

Workplace support is a prerequisite for improving work ability. In this study, we analyzed papers from 1978 to 2020. This is the period during which the treatment of mental illnesses changed, from focusing on drug-related treatments to social participation [49].

Based on this concept of mental illness, we believe that the requirements for work ability include not only individual factors but also content that is related to the work environment, which provides support for people with mental disorders. From the perspective of improving the work environment, it is necessary to consider the following three points: ① building human relationships in the work environment, ② creating a supportive environment for the entire workplace, and ③ integrating the workplace into society. These will be explained below:

① Building human relationships in the work environment

Creating a productive work environment requires organizational efforts to build relationships among workers and between workers and consultants, such as by improving relationships among workers and establishing a system of consultation support for them. This is similar to the organizational approach of daycare, which provides rehabilitation for people with mental disorders, toward the goal of social participation [50]. It is necessary for others (i.e., coworkers, family members, friends, acquaintances, managers, etc.) to understand the importance of human relations at work, to facilitate the healing process for people who are recovering from mental health difficulties. Enhancing support for human relations in the workplace will create an environment that is conducive to employment, including opportunities for workers to share their mental health and challenges of employment and human relations, as well as building caring human relations when returning to work, which in turn will improve their ability to work.

② Creating a supportive environment for the entire workplace

In addition to the creation of a work environment that values relationships that can develop working abilities, it is necessary to create a supportive system for the entire workplace. This study has shown that the ability to work can be improved in stages. To promote this, it is necessary to have an employee role dedicated to coordinating support that is tailored to the situation of the worker within the workplace. Coordination is a process of interaction between participants, and human relationships play a large role in problem solving [51]. Therefore, this coordinator needs to understand the importance of human and workplace relationships. The coordinator should play a role in sharing the goals of employment support for the worker and the workplace, as well as issues of implementation, building mutually respectful relationships within the workplace, and developing consultation and support systems at the time of entry (or return) to the workplace. By promoting these activities, the coordinator will put the entire workplace system into practice. For a long time, countries have adopted the EAP (employee assistance program). This consists not only of individual productivity but also organizational care, such as self-care by the disabled themselves, line care by managers, and care by staff inside and outside the office. EAP have been proven to be feasible and effective, with relatively short involvement [52,53,54]. Therefore, we believe that proactive implementation of EAP can solve the challenges to this problem.

③ Integrating the workplace into society

While people with mental disabilities are shifting to a trend of speaking openly, they may also be subject to prejudice against such behavior. Therefore, as many campaigns have freed people from the suffering of mental disorders [55], it is important to encourage society to act with consideration for people with mental disorders who are trying to return to work and to create a workplace culture that improves their ability to work.

Thus, based on the results of the present study, we believe that to improve the work ability attributes of people with mental disabilities, there should be a limit to the prior requirements of individuals. It is important to create and foster a supportive work environment where people with mental disabilities and society as a whole can belong, which encompasses the workplace.

Even after a temporary suspension, it is important to determine its cause and continue to develop the required skills to achieve the desired consequences (Figure 2).

### 4.2. The Role of Nursing in Supporting Work Ability of Persons with Mental Illnesses

Nursing that supports the work abilities of persons with mental illnesses is triune in function: ① it supports the improvement of health management and basic daily living; ② it encourages interpersonal interaction and active orientation toward work; and ③ it prepares a follow-up system for multidisciplinary collaboration and employment. Nurse involvement is crucial at the start of the intervention, and when employees’ symptoms become unstable during employment. Such support plays a significant role in strengthening antecedent conditions.

① Support for the improvement of health management and basic daily living

The stabilization of symptoms in the work environment is an imperative condition for people with mental illnesses. Thus, nurses should be on-hand to provide daily support and medical treatment to patients prior to or during work if their symptoms become unstable, during hospitalization, or in any scenario.

The focal antecedent conditions at the start of the intervention are the ability to manage health and the ability to manage basic daily living. Therefore, nurses must initially assist with the tasks in the work process (ADL, symptom management, making judgments, etc.) while encouraging patients to affirm their ability to care for themselves. For example, nurses could alert the patient when it is time to eat or sleep, thus improving their rhythm of life and strengthening their physical sensations. Furthermore, nurses can attempt to improve the patient’s ability to manage basic daily living to balance activities and rest while gradually grasping the flow of activities that constitute their work routine. Regarding the ability to manage health, nurses can check the details of the psychiatric drug therapy, through psychoeducation and individual consultations, and consider their effective use during and after work hours together with the patient.

During the examination of an employee patient, the examining nurse must treat the main symptoms and the functional impairment that the illness causes. They must also inform the patient about the effects of the mental illness on work life, family life, and their relationship with coworkers while improving their problem-solving abilities. This involvement accompanies individual support and crisis intervention as elements of the ability to manage health. Rehabilitation includes elements of the ability to manage basic daily living, such as control of their rhythm of life to facilitate employment, as well as time-allocated leisure time.

Therefore, the role of nurses at the start of an intervention prior to and during employment is to encourage the recalibration of their patient’s rhythm of life, the effective use of medication, and a balanced life outside of work.

② Encourage interpersonal interaction and active orientation toward work

As symptoms stabilize prior to employment, the next step is to accumulate meaningful experiences and effective interactions with others, to acquire the required antecedents (i.e., skills for interpersonal interaction and active orientation toward work). As people with mental illnesses may encounter difficulties when building interpersonal relationships, it is important to improve their skills for interpersonal interaction that inform their adaptability, a necessary skill that favors continuous employment. Therefore, nurses should be aware of events that take place outside of the patient’s employment while observing symptoms. At the stage of hospitalization or outpatient treatment, nurses must also provide support to enable active interactions with people outside of their immediate entourage. To date, stress in life at home and in society has greatly affected the adaption of mental patients after their discharge. As such, nurses have incorporated social skills training to develop patients’ interpersonal interactions. Thus, nurses can provide support to improve communication skills, as well as education on psychiatric drugs that are necessary for their patients’ employment experience. We encourage the educational involvement of nurses in the framework of social skills training, which involves instructing the patient on how to request assistance when in need or offering assistance to another person [15].

Furthermore, motivation and active orientation toward work are imperative qualities for people with mental illnesses to find employment or continue working. Interpersonal interaction is important, but professional goals should be set, and the motivation to accomplish objectives related to those goals should be demonstrated and maintained [28,34]. For that purpose, the patients need to utilize their goals actively, as well as their own knowledge and skills.

Nurses can provide motivation and positive feedback to help them aim for independence if they are currently undergoing life training and medical treatment at available facilities. When practicing active orientation, they can employ a strength model to visualize the patient’s strengths and progression with their condition thus far. Since recovery is not a linear process, it is necessary to encourage patients in a manner that makes them aware of the small breakthroughs on their path toward recovery. Nurses can improve active orientation toward work by providing positive feedback that considers the strengths of patients, which they can apply at work, and which assures them of their gradual improvement.

③ Establishing multidisciplinary collaboration and an employment follow-up system

In addition to individual antecedents, workplace antecedents (i.e., the workplace that believes in employees’ ability to work, the workplace that has a support system, and the workplace with a sense of security) are important when symptoms become unstable while working. Work content and environment require adjustments according to the medical condition of people with mental illnesses; thus, it is necessary to provide integrated support involving professionals from related institutions surrounding the patient. If a hospitalized mental patient desires to start earning a living, the ward nurses must collect information (e.g., current goals, previous work experience, or the support status at their workplace) and share these with other nurses and managers at the prospective workplace.

Ward nurses can participate in the patient’s education through group psychotherapy and individual consultation regarding medication management and daily life, with the patient’s goals in mind. Occupational health nurses can gather information from the patient’s coworkers and superiors about the patient’s work environment, as well as the patient’s ability to seek help voluntarily. With this information, nurses can recommend methods to reduce their workload and advise workplace managers. This will, in turn, allow patients to share emotions, focus on their consultations, and search for possibilities based on their needs, to allow progression on their own terms. In other words, we believe that nurses can perform support activities for the development of follow-up systems for persons with mental illnesses and those around them. In addition, outpatient nurses and visiting nurses can share information with the doctors or mental health workers. They are able to adjust the patient’s rhythm of life, check their medication status, and provide direct support at home. A family member can forward information to the nurse, thus supporting the improvement of the patient’s home life during their employment.

In the UK, employment advisors provide decision-making support for adjustment in the workplace, leading to more effective job-hunting [56]. Expressly, the collaboration of medical care (doctors, outpatient nurses, visiting nurses, occupational health nurses, employment advisors, etc.) and specialized support in fulfillment of the desired outcomes may enable employment follow-up according to the person’s medical condition. This could positively influence effective employment and productivity for persons with mental illnesses. As described in ①, ②, and ③, seamless collaboration is necessary when considering the individual roles of the associated practitioners. Thus, mental patients tend to be more productive and function more efficiently when they have access to medical care that can provide comprehensive support for medical care, work, and daily life. Individual and workplace antecedents are at the very foundation of work ability. We believe that the role of nurses is to provide specialized support, build a multidisciplinary follow-up system to help mental patients improve their work skills, and reflect on fundamental antecedents.

## 5. Conclusions

This study analyzes the concept of the work ability of people with mental illnesses, based on Rodgers’ conceptual analysis. The attributes of work ability include the ability to self-manage, adaptability, the ability to dedicate oneself to work, and the ability to formulate plans, which are developed through a reiterative process. We divided the antecedents of these attributes into individual and workplace antecedents. Workplace antecedents include a workplace that believes in employees’ abilities to work, possesses a support system, and possesses a sense of security. On the other hand, individual antecedents include skills for interpersonal communication, active orientation toward work, the ability to manage health, and the ability to manage basic daily living. The consequences include attaining independence, developing interpersonal relationships, improvement and maintenance of self-care, achieving happiness, and positive self-transformation.

It is important to adjust the content and environment of work according to the employee’s condition. Therefore, the ability to cope with stress, workload, and active self-adjustment are foundational skills required of persons with mental illnesses. In sum, to support the work abilities of persons with mental illnesses, nurses must assess the person’s daily life, aid in improving the skills necessary for continuous work, and forge multidisciplinary collaborations and a follow-up system after employment.

### Limitations

The required ability to work changes according to the culture and policies of the country and its historical background. However, this study did not analyze the differences of the concept among countries, because it clarified the overall concept of work ability by analyzing the comprehensive descriptions of work in each paper. To use this concept in practice, it is necessary to clarify the characteristics of the ability to work according to the culture and policies of each country. Therefore, not analyzing the concept by country is a limitation of this study.

## Figures and Tables

**Figure 1 ijerph-18-10172-f001:**
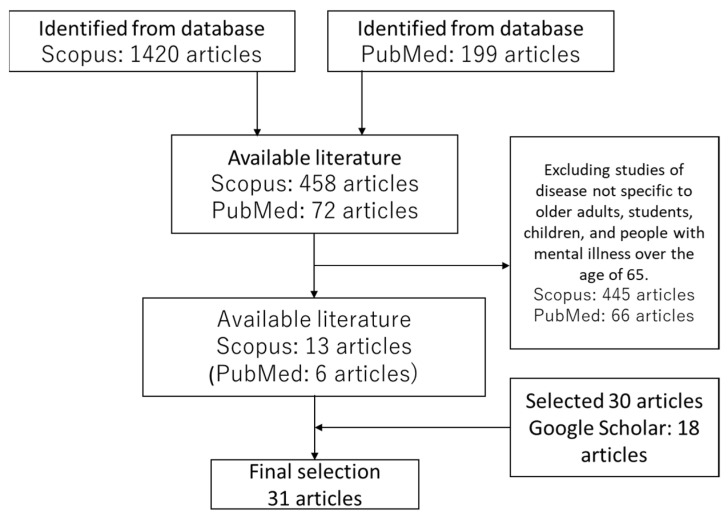
Flowchart describing the target literature search for final selection.

**Figure 2 ijerph-18-10172-f002:**
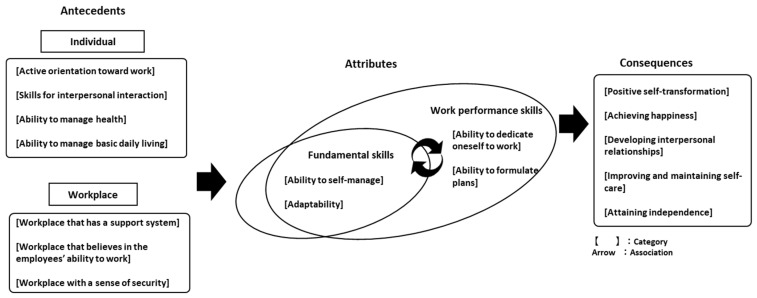
Conceptual diagram of the work ability of people with mental illnesses.

## Data Availability

No new data were created or analyzed in this study. Data sharing is not applicable to this article.

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
