# Peer review of "The Work Ability of People with Mental Illnesses: A Conceptual Analysis"

_ijerph, 2021, doi:10.3390/ijerph181910172_

Round 1
Reviewer 1 Report
Section 3 results: Why didn't you use the category Employee Assistance Program (EAP)?
Section 4 Discussion: I suggest to deepen organizational-level interventions could be the management of the organization, the leadership
Reviewer 2 Report
I agree that continuous employment is an important goal for many people with mental illness. The aim of this study to analyze the conceptual framework of the work ability of people with mental illness. The good point of view is to emphasize a holistic approach rather than the treatment of symptoms.
Strengths of the manuscript:
- The authors used Rodgers' evolutionary conceptual analysis. This qualitative and inductive method is applicable to the examination of the work ability of persons with mental disorders, which is greatly dependent on historical, cultural, and ethical contexts.
- The search yielded 1,420 articles in Scopus and 199 articles in PubMed. The study targeted literature published from 1978 to 2020, in order to visualize the concept of “work ability of persons with mental disorders.” The academic fields used for data collection, included nursing, psychiatry, sociology, welfare, and psychology, while the searched inventories included Scopus, PubMed, and Google Scholar. The authors searched for literature bearing the words “mental,” “employment/work,” “ability,” “motivation,” and “performance” as keywords.
- During their analysis, raw data about the attributes, antecedents, and consequences that appeared to correspond to their context, were extracted from each piece of literature. The data were labeled and codes were created. They created names to indicate sub-categories while examining any disparities among the codes. Any further abstracted codes were considered as categories.
- Authors discussed the results and how they can be interpreted from the perspective of previous studies and of the working hypotheses.
Results: Authors analyzed 31 articles in this study. Depending on the field of the journal, they assigned the pieces of literature to the following five categories: mental health (n =2), rehabilitation (n =6), psychology (n =8), occupational medicine (n =5), and psychiatry (n =10). The conceptual analysis yielded 4 attributes, 7 antecedents, and 5 consequences of the “work ability of people with mental illness.”
Conclusions.
- This study analyzes the concept of the work ability of people with mental illness, based on Rodgers' conceptual analysis. The attributes of work ability include the ability to self-manage, the adaptability, the ability to dedicate oneself to work, and the ability to formulate plans, which are developed through a reiterative process.
- Workplace antecedents include the workplace that believes in the employees’ ability to work, the workplace that has a support system, and the workplace with a sense of security.
- On the other hand, individual antecedents include skills for interpersonal communication, active orientation toward work, ability to manage health, and ability to manage basic daily living. The consequences include attaining independence, developing interpersonal relationships, improvement and maintenance of self-care, achieving happiness, and positive selftransformation.
- To support the work abilities of persons with mental illness, nurses must assess the person’s daily life, provide assistance toward improving the skills necessary for continuous work, and forge multidisciplinary collaborations and a follow-up system after employment.
Conclusions: The article written in an appropriate way. The results interpreted appropriately. Article is suitable for publishing.
Reviewer 3 Report
Thank you very much for giving me the opportunity to review this paper. I feel that the focus of this research will be very useful in supporting the employment of patients with mental illness, and that it may also be useful in employment measures for healthy people. I would like the authors to modify and release this paper to the world.
In this study, authors used a technique called conceptual analysis to summarize the concept of “work ability” over 40 years of people with mental illness, and also mention the support system by medical staff. As a result, it would be great if the classified and organized concept of “work ability” could be used to support the employment of people with mental illness.
However, I think the following changes are necessary.
Major comments:
- L,293~
In discussion, authors should discuss about a logically constructed hypothesis from the results of this study. Since only few reference is cited in this 3 pages, it seems as if the author's theory is stated about the way of nurse‘s work. Although it is meaningful in clinical situation, I don't think it fits scientific papers.
If you write this paragraph(L210~213) in this paper, “Authors should discuss the results and how they can be interpreted from the perspective of previous studies and of the working hypotheses. The findings and their implications should be discussed in the broadest context possible. Future research directions may also be highlighted.”, please write the content exactly as you declared.
2, The limitations of this study have not been examined or described. I think it's usually mentioned.
- Please correct the some misspellings and check the grammar problems
- I think it would be better to discuss about the future outlook or problems to be improved from an overall perspective rather than from the nurse's point of view, since the readers of this journal seem to be in various occupations.
Minor comments:
- L5 The description of the author's department is missing. It is necessary to add.
- L35 Is reference 5 from the American Psychiatric Nursing Association?
- L57 “In Japan, there are structures to develop basic skills, such as critical thinking, for working adults.”
Is there any reference related to this description?
- L98 The description of Figure.1 is duplicated.
- L99 Results
Authors do not mentioned the countries where the selected final 31 studies were performed. Is there any difference in concept between countries?
- L132 What does” [“ and “] “mean?
- L210~213 This paragraph is inappropriate.
- L335 Is there any reference about the SST description?
Reviewer 4 Report
This is an interesting article in which the authors investigate the work ability of people with mental illness using Rodgers' (2013) evolutionary conceptual framework. It would have been useful to explain why alternative conceptual frameworks were not chosen. There is a clear and compassionate rationale for this study with a constructive approach that potentially should help healthcare professionals to support people with mental health difficulties as they make the transition back to work. The authors systematically searched the existing literature over a long timespan and narrowed down the selected articles to 31. The authors give a detailed description of the processes that they used to code the findings, as summarized in Figure 1, from which they elicited four main attributes of work ability: self-management, adaptability, dedication to work, and the ability to form plans. In addition to the attributes of the individuals concerned, the authors also take some account of the conditions in the workplace. This is important since, without a support system and a commitment to providing a sense of security to employees, the rehabilitation process would be much harder for the individual in recovery from a mental illness. These systems also need to be backed up by a legal framework if stigma and discrimination are to be challenged.
I wondered if the authors might have included more discussion of the capacity of people with mental health difficulties to work harmoniously with others and to communicate constructively with colleagues. This is mentioned under the heading of "Antecedents of work ability" but is not reflected in the four attributes. Some discussion of the potential links between the antecedents and the attributes elicited from the literature would have been helpful. However, the authors do make the important point that there is an urgent need for healthcare professionals and (by implication) managers in the workplace, to be made aware of recent research findings in order to guide their practice in the treatment and rehabilitation of persons with mental health difficulties. The authors of the present article place great emphasis on the individual's skills, capacity to formulate a plan, and capacity to self-manage in their discussion. In my view, this should be balanced by an emphasis on the humanity and sensitivity of management in the workplace, with systems in place to provide support and guidance as the individual re-establishes themselves back into work. The conclusion mentions the importance of multidisciplinary collaboration which, in my view, is a key component. Since the authors of the present study selected research studies from a large timespan, perhaps they could have taken more account of changing attitudes towards mental illness in society at large over time and the influence of such campaigns as World Mental Health Day and the Anti-Stigma Campaign. The authors could also have mentioned the value of heightening the awareness of colleagues and helping them to act in a sensitive way towards others who have suffered a mental health issue but who are trying to return to the workplace. In other words, the authors could have moved beyond a statement of the attributes that they elicited from the research studies to a more fundamental critique of what they discovered.
I would recommend Foulkes, L. (2021). Losing Our Minds. London: The Bodley Head for a challenging overview of the current conceptualization of mental illness.
I suggest that before the article is accepted, the authors do the following:
- Provide a more detailed rationale for choosing Rodgers' conceptual framework instead of others;
- Critique the findings more forcefully bearing in mind that attitudes to mental illness have changed over the time span of the literature that they reviewed;
- Even though the findings focussed on individual attributes, place more emphasis on the social context of the workplace, to include the policies concerning employees with a mental health issue, the existence or not of training for managers and colleagues in working in a caring and sensitive way with a person who is returning to the workplace after some form of mental health difficulty;
- Show awareness of current conceptions of mental illness and controversies in the field;
- Emphasize the need for others (whether colleagues, family, friends, acquaintances, managers) to give the person in recovery from a mental health difficulty the time and space to get better, to provide awareness of the need to understand that there will be troughs and peaks involved in the process, and to appreciate the importance of relationships in the workplace in facilitating the healing process.
Round 2
Reviewer 3 Report
Minor comments:
L.315
Please write the non-abbreviated word when the abbreviation EAP first appears.
Reviewer 4 Report
Please thank the author for revising the article in light of my comments. I am happy now for it to be accepted for the journal.
I noticed one small grammatical issue that needs to be corrected. One of the sentences in the section on EAP is not a full sentence so should be changed as follows:.
EAP have been proven to be feasible and
effective with relatively short involvement [4–6]. Therefore, we believe that proactive implementation of EAP can solve the challenges to this problem.
